
# Parallel assembly of neutral atom arrays with an SLM using linear phase interpolation

Ivo H. A. Knottnerus[1,2,3], Yu Chih Tseng (曾于誌)[1,2], Alexander Urech[1,2], Robert J. C. Spreeuw[1,2] and Florian Schreck[1,2]⋆

1 Van der Waals-Zeeman Institute, Institute of Physics, University of Amsterdam, Science Park 904, 1098XH Amsterdam, The Netherlands
2 QuSoft, Science Park 123, 1098XG Amsterdam, The Netherlands
3 Eindhoven University of Technology, P.O. Box 513, 5600MB Eindhoven, The Netherlands

⋆ slmsorting@strontiumbec.com

## Abstract

We present fast parallel rearrangement of single atoms in optical tweezers into arbitrary geometries by updating holograms displayed by an ultra fast spatial light modulator. Using linear interpolation of the tweezer position and the optical phase between the start and end arrays, we can calculate and display holograms every few ms, limited by technology. To show the versatility of our method, we sort the same atomic sample into multiple geometries with success probabilities of 0.996(2) per rearrangement cycle. This makes the method a useful tool for rearranging large atom arrays for quantum computation and quantum simulation.


# 1 Introduction

Large arrays of neutral atoms are proving to be a quintessential tool for quantum simulation [1–4] and quantum computation [5–7]. Unlike on many other platforms, scaling up the number of physical qubits in a neutral atom tweezer machine is often a matter of enough laser power, and arrays of up to thousands of single atoms have been demonstrated [8–12]. Recent work explored continuous loading of neutral atom arrays in lattices to increase this number even further [9, 13, 14].

To harness the full potential of many interacting single atoms, rearrangement of atoms is necessary to create defect-free geometries. For quantum computing, rearrangement is also used to shuttle atoms in between storage, interaction and imaging zones [8, 10]. Most often, static patterns are made using a spatial light modulator (SLM) [5, 8] or lattices [9, 14], but the rearrangement of atoms is almost exclusively done by acousto-optic deflectors (AODs), because of the fast response time of AODs compared to an SLM [15]. However, simultaneous sorting of more than one row of atoms requires multiple AODs and is limited in the types of sorting moves that can be implemented [5, 16]. Furthermore, moving an atom with an AOD while using an SLM for static tweezers adds a handover from SLM to AOD tweezers and back per move, which requires a high degree of alignment and optimization, introduces a non-zero loss probability and takes constant overhead time [5, 17].

A way to circumvent these problems is rearrangement with an SLM. In previous work [18, 19], a modified weighted Gerchberg-Saxton (WGS) algorithm was implemented to update holograms on a high-speed SLM, moving atoms in parallel to non-lattice geometries [19–22]. However, lack of computational strength resulted in slow rearrangement cycles. In the meantime, significant atom losses per movement imposed the need for multiple rearrangement cycles to obtain defect-free arrays. In combination with a limited vacuum lifetime, these problems made the method so far unpractical for large scale quantum simulations and computations.

In this article, we present a fast and efficient method for parallel atom rearrangement with an SLM. We demonstrate a linear phase interpolation method, controlling not only the position, but also the optical phase of tweezers in successive holograms. The control of the optical phase reduces intensity flicker of tweezers during moves and thus enhances the probability that an atom survives these moves. Fixing both the amplitude and the phase of the desired tweezer geometry also greatly reduces computational complexity and allows us to reach computation and display cycles of $2.736(6)$ ms on commercial hardware.

In the first part of the paper, we present experimental results of testing our method on a $6 \times 6$ tweezer array. This results in transport losses that are competitive with AODs, with success probabilities of $0.996(2)$ per tweezer per rearrangement cycle. In Sec. 3, we present a detailed investigation on how controlling the optical phase of the tweezers minimizes the atom loss. In Sec. 4, we present benchmarks of the computational time and show that it is nearly independent of the number of tweezers. Finally, we present an outlook on potential use-cases of the developed technique in Sec. 5.

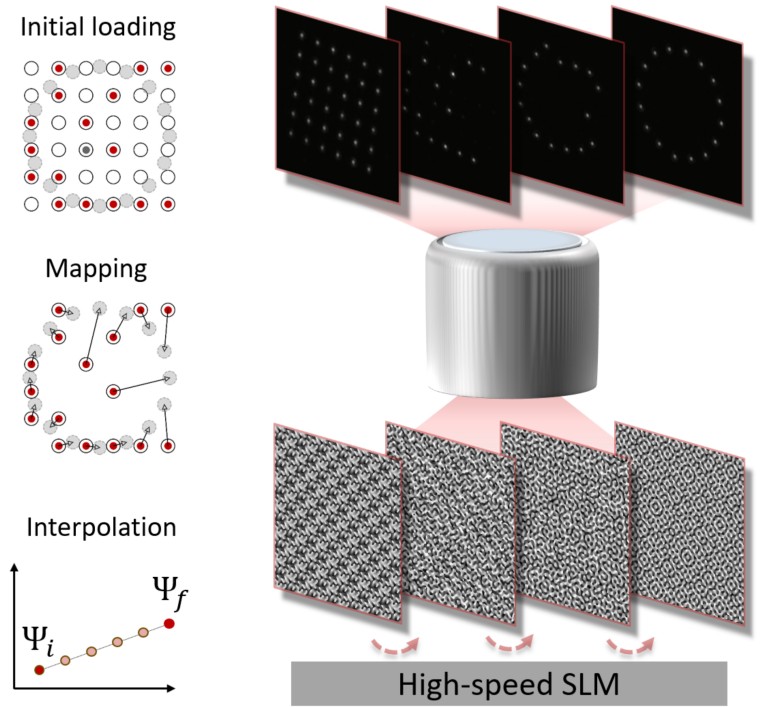

Figure 1: Schematic overview of our experimental sequence for fully parallel rearrangement into arbitrary geometries. At the start of the sequence, single atoms are stochastically loaded into an initial tweezer geometry with approximately 45 % probability per tweezer. Empty tweezers and tweezers with excess atoms are extinguished. Remaining atoms are mapped to target positions and trajectories are calculated that move the other atoms into the desired geometry. Using linear interpolation of position and optical phase of the tweezers, the trajectories are divided into multiple steps. For each step a hologram is calculated in real time and displayed on a high-speed SLM, moving all atoms at the same time.

## 2 Parallel rearrangement of atoms with SLM

In this section, we present the experimental realization of fully parallel rearrangement of single $^{88}$Sr atoms using an SLM. The experimental apparatus is largely the same as the one used in references [23,24], with the main difference being the installation of an ultra-high speed SLM (Meadowlark UHSP1K-488-850-PC8) with over 1 kHz refresh rate on the optical path to generate the 813-nm tweezer patterns, see Appendix A. In Fig. 1, a schematic of the experimental sequence is presented. At the start of each experimental cycle, a single atom is prepared in about 45 % of the tweezers and detected with 99 % imaging survival. A set of trajectories is calculated to link atoms from their starting tweezer to the desired final location. The trajectories are divided into multiple steps by linearly interpolating both the position and the optical phase of every tweezer from start to end. For each step, a hologram is calculated and displayed on the SLM so that all atoms move in parallel from their starting position to their final position. The unused tweezers are ramped off before any move. Details of the method used to calculate the holograms are presented in Sec. 3. After the atoms have been rearranged, a verification image is taken.

In Fig. 2a, an example experimental realization of sorting a 6 × 6 tweezer array into a defect-free 4 × 4 tweezer array is presented. In both patterns, the atoms are spaced 6 μm apart. From the 20 atoms loaded initially (left panel, green circles), 16 atoms are selected

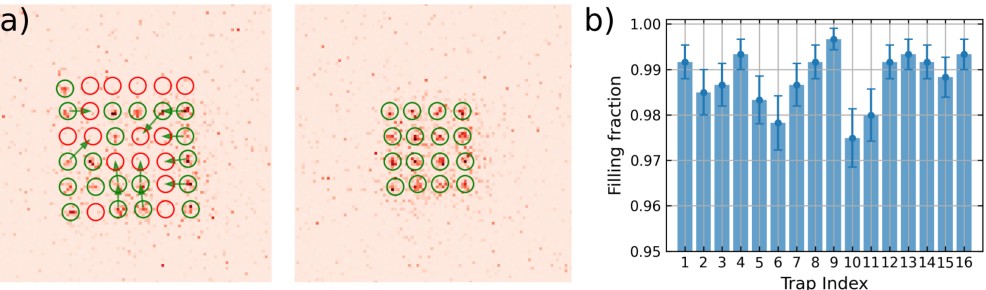

Figure 2: (a): Fluorescence images of single atoms stochastically loaded into a square $6 \times 6$ array (left panel). The green (red) circles denote the presence (absence) of an atom in a tweezer. The arrows show trajectories that sort 16 atoms into a defect-free $4 \times 4$ array. The right panel shows the verification image after rearrangement. (b): The average filling fraction per tweezer in the verification image for 1000 experimental realizations. The error bars are the standard deviation of the mean. The average filling fraction over all tweezers is 0.988(4). Corrected for the image survival, this corresponds to a rearrangement success of $0.997^{+0.003}_{-0.006}$ per atom.

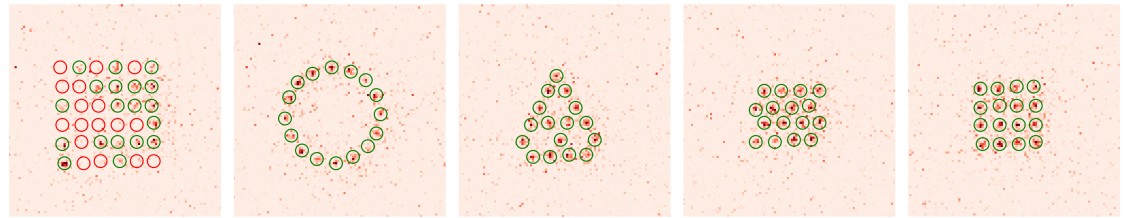

Figure 3: An example experimental realization of the same atoms being sequentially rearranged to various geometries. From left to right: image of the initially loaded $6 \times 6$ array, a circular geometry, a kagome lattice, a triangular lattice and a $4 \times 4$ array.

and rearranged to the final geometry (right panel, green circles). To characterize the loss of the method, the experiment was repeated 1200 times. In Fig. 2b, we plot the probability of detecting an atom in each tweezer in the verification image, under the condition that there were 16 or more atoms detected in the initial image. The mean probability is 0.988(4), with a minimum probability of 0.975(6). The error bars represent the standard deviation of the sample. The detection probability is mainly limited by the survival rate of the atoms during the first image. Correcting for this survival, we obtain a success probability of $0.997^{+0.003}_{-0.006}$ per atom for the rearrangement, see Appendix B. For the particular instance presented in Fig. 2a, the rearrangement was performed using 13 hologram updates, which took in total around 40 ms.

A benefit of using an SLM over using crossed AODs for rearrangement is the ability to easily change geometries during experiments. This allows for complex connectivity changes within quantum simulations or computations. We experimentally investigated the ability of our method to sequentially change geometries, by arranging 16 atoms out of the initial $6 \times 6$ geometry into a circle, a section of a kagome lattice, a triangular lattice, and finally a $4 \times 4$ square grid. Fig. 3 shows images taken directly after loading and after each rearrangement. The spacing in between neighboring atoms was the same in all geometries. Repeating this experiment 2000 times while monitoring the filling fraction in the final image yielded an average filling fraction of 0.968(7). Assuming the same success probability for each rearrangement and correcting for losses induced by imaging in the different geometries, this results in a rearrangement success of 0.996(2) per atom per cycle.

## 3   Linear phase interpolation for flicker-free transport

Next, we go into details of the method used for calculating the holograms for the SLM. In previous work, all holograms for rearrangement were calculated using the WGS algorithm [19, 25]. The WGS algorithm leaves the optical phase of the tweezers as optimization parameters to iterate towards the desired amplitude distribution. As a consequence, if we use the WGS algorithm to calculate a sequence of holograms for atom rearrangement, the optical phase of the tweezers can jump randomly from hologram to hologram. This leads to intensity flicker during the switching transient from one hologram to the next, which in turn leads to heating and potential loss of atoms. To overcome this issue, we propose and implement a linear phase interpolation method (LPI) for calculating a sequence of holograms for atom rearrangement.

As a starting point of our method, we first calculate holograms for the initial and final tweezer pattern using the WGS algorithm [19, 25]. To minimize imaging losses in the experiment, we typically perform several iterations on both the initial and final patterns to reach a tweezer trap depth deviation of less than 1 % from the average trap depth [24, 26]. The trap depth deviation is measured with spectroscopy on the narrow $^1S_0-^3P_1$ transition of strontium. By taking the fast Fourier transform (FFT) of the initial hologram, we obtain for each tweezer a position, an amplitude, and an optical phase. We do the same for the final hologram.

Each rearrangement cycle starts with stochastically loading atoms into the initial geometry. Using the Jonker-Volgenant algorithm, we map loaded atoms to target tweezers in the final geometry while minimizing the maximum length of trajectories by using a squared distance cost function [27, 28]. After determining which tweezers will be used in the rearrangement, all other tweezers are ramped off by computing and displaying a low number (typically one or two) of holograms that linearly ramp down the amplitude of the unused tweezers and ramp those of the remaining tweezers to the values in the final hologram, while keeping the positions and optical phases of all tweezers fixed. The tweezer amplitudes are normalized, which results in a redistribution of the light power that was used for the extinguished tweezers onto the remaining tweezers. For the remaining tweezers, we linearly interpolate the position and the optical phase between their initial and final values over $N$ steps. For each step, a hologram is calculated by taking the inverse FFT of the interpolated tweezer pattern. By performing an FFT on a commercially available high-end GPU (Nvidia GeForce RTX 4090) with OpenCL and VkFFT, the total computational time of a single hologram for the 1024 × 1024 pixel SLM takes less than 200 µs [29]. When the atoms have arrived in the final geometry, the hologram is identical to the intensity balanced final hologram.

The total number of intermediate holograms $N$ is chosen to match the longest trajectory in Fourier units of $\frac{\lambda f}{mL} \approx 0.45\,\mu\text{m}$, where $\lambda$ is the laser wavelength, $f$ the focal length of the objective and $mL$ the demagnified side length of the square SLM chip with $m \approx 0.41$. In this way, atoms move at most one Fourier unit per hologram in both directions, which ensures an overlap between consecutive tweezers with a Gaussian beam waist of 0.88 µm, even for diagonal moves.

The effect of the phase restriction on the intensity flicker of optical tweezers during transport is characterized on a separate test setup using a high-speed camera (Phantom Miro 110) with 11-kHz frame rate. In Fig. 4a, we compare the relative intensity flicker of a single moving spot generated using the WGS algorithm with a spot of which the optical phase remains the same throughout all holograms (LPI for the special case of identical initial and final tweezer phase). While the intensity of the LPI-spot never drops more than 30 %, the WGS-spot almost vanishes completely several times. As an explanation, we calculate the FFT of each hologram and plot in Fig. 4b the expected optical phase at the tweezer location in each frame. The optical phase of the WGS-generated spot varies strongly over the different patterns. During the update of the SLM we observe transient patterns on the camera, displaying spots at both the

original position of the tweezer and at the new position. When those (partially overlapping) spots have a large optical phase difference, destructive interference creates intensity flicker.

To determine the maximally allowed phase slip per hologram for atoms to survive, we load atoms in a $6 \times 6$ tweezer array and translate all the spots five steps of one Fourier unit in one direction and then reverse this movement. During each step, we programme a variable slip $\Delta\psi$ of the optical phase of the tweezer. When reversing the movement, we flip the sign of this phase slip. We image the atoms again after they have returned to the original position. We plot the survival as blue disks in Fig. 5a. It can be seen that tweezer phase slips of up to several tenths of $\pi$ radians per step do not drastically influence the atom survival, whereas phase slips around $\pi$ lead to atom loss.

A moving tweezer can also acquire a phase slip due to an unwanted displacement $d$ of the real optical axis and the optical axis assumed when calculating holograms. Moving a tweezer sideways corresponds to adding a phase gradient to the hologram, equivalent to the phase change induced by a mirror at the SLM location that is being rotated to shift the tweezer position. If the gradient's origin (i.e. the "mirror's rotation axis" or the "computational center" of the hologram, see also Appendix C) coincides with the optical axis, the center of the beam has the same phase for every gradient. Consequently, changing the phase gradient does not change the tweezer phase, see Fig. 5b. However, if there is a displacement, the tweezer's phase experiences a phase change $\xi$ in addition to the desired $\Delta\psi$ in each movement step, see Fig. 5c. In Appendix C we show that $\xi$ depends on the displacement $d$ (given in SLM pixels) as $\xi(d) = 2\pi d/M$, where $M$ is the total number of SLM pixels along one dimension. Tweezer motion can lead to atom loss if the total phase change $\Delta\psi + \xi$ approaches $\pi$.

To probe the effect of misalignment experimentally, we perform the same phase-slip experiment as described above, but with the computational center of the hologram displaced from the optical axis. The average survival rate of the atoms is given by orange triangles in Fig. 5a. The shift of the computational center is 250 pixels, which for our 1024 pixel wide SLM leads to a shift of $\xi \approx 0.5\pi$ per step. This is in complete agreement with the observed shift of the maximum atom loss by $-0.5\pi$ in $\Delta\psi$. Note that this type of measurement can be used to determine the displacement between computational center and optical axis, thereby providing the information needed to null this displacement.

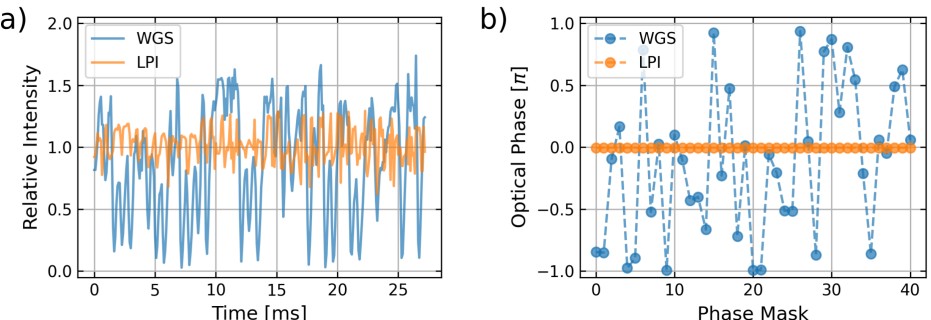

Figure 4: (a): The relative intensity recorded with an 11-kHz frames-per-second camera of a moving spot made by holograms generated with the WGS algorithm (blue trace) and made with LPI for the special case of identical initial and final tweezer phase (orange trace). The WGS-generated spot has multiple dips with almost no intensity left, while the LPI-generated spot stays above 70 % throughout the moves. (b): The calculated optical phase at the location of the moving spots for each hologram displayed during the movement.

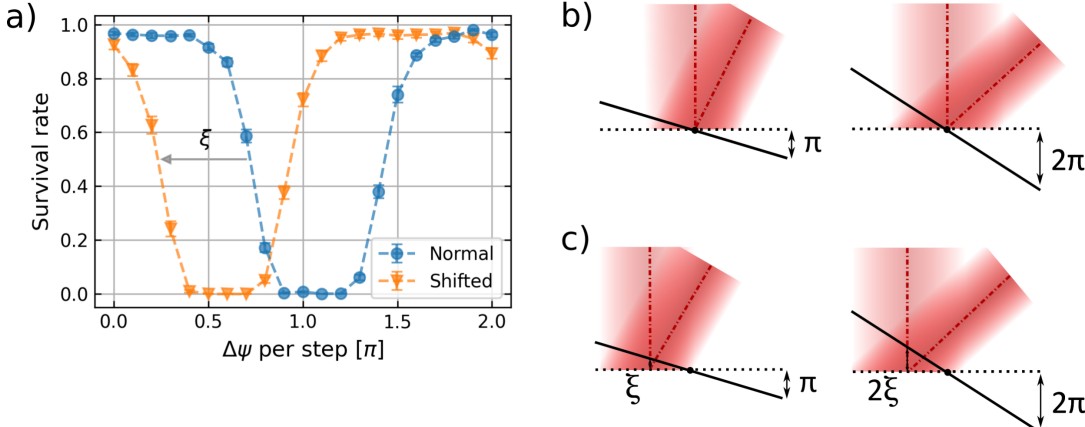

Figure 5: (a): The average survival rate of atoms in a $6 \times 6$ array after having moved 10 steps of one Fourier unit in total with a tweezer phase slip of $\Delta\psi$ per step. The blue trace is the result while taking as computational center (see Appendix A) the center pixel of the SLM, which leads to a maximum loss at $\Delta\psi \approx 1.05\,\pi$. To measure the effect of alignment, a second scan was performed (orange trace) with a shift in computational center of 250 pixels. This results in an additional tweezer phase slip per step of $\xi \approx 0.5\,\pi$. Dashed lines are a guide to the eye. (b) and (c): Translating a spot using the SLM is equivalent to changing the slope of a phase gradient hologram (solid line) displayed on the SLM (dashed line). The left and right panels correspond to consecutive holograms moving the tweezer by one Fourier unit, which increases the hologram phase value at the edge of the SLM by $\pi$. When the optical axis (dash dotted line at center of laser beams) crosses the SLM at the position at which the phase gradient crosses zero (black dot), no phase shift is acquired by a tweezer when moving it by one Fourier unit, as shown in (b). When these positions do not match, each translation introduces an additional tweezer phase shift $\xi$, as shown in (c). The reflection angles of the outgoing beams are not to scale.

## 4 Scaling to large atom arrays

Due to the fully parallel nature of the transport, SLM rearrangement has the promise of scaling very well to large-number atom arrays. Where rearrangement with only AODs is limited to moving at most a row of atoms per AOD at a time, with the SLM all atoms can move simultaneously, in individual directions. If SLM-created tweezer patterns are required for the experiment at hand, using the SLM also for sorting furthermore alleviates the atom loss and overhead time associated with transferring atoms from SLM tweezers to AOD tweezers and back.

For the LPI method, the computational time needed to determine updated tweezer positions and phases for the next hologram in a rearrangement move scales linearly with the total number of tweezers $N_{tw}$. Modern hardware executes these operations in nanoseconds even for thousands of tweezers, which is negligible on the scale of the SLM refresh time. Fourier transform and SLM refresh time are independent of the number of tweezers, which means that the SLM update cycle time is effectively independent of $N_{tw}$.

We benchmark the timing for our method by moving a $\sqrt{N_{tw}} \times \sqrt{N_{tw}}$ sized array 10 steps for 10000 repetitions, with $N_{tw}$ varying from $3^2$ to $49^2$. The measurement was repeated twice in ascending order, and twice in descending order. During the process, we monitor: the time it takes the CPU to calculate the next tweezer positions and phases; the time it takes the GPU to update the buffers and calculate the next hologram; the time it takes to transfer the hologram

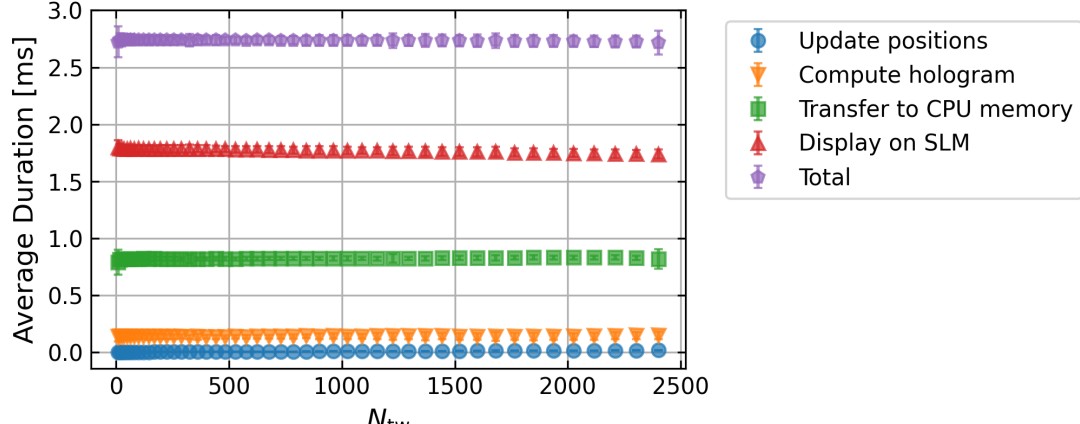

Figure 6: Independence of the average duration per hologram of different parts of the method for increasing number of tweezers $N_{tw}$. The purple pentagons denote the total duration of a computation and display cycle, which consists of updating of the tweezer positions (blue circles), computing the next hologram on a GPU (orange downward triangles), the transfer of this hologram from GPU to CPU memory (green squares), and the display of the hologram on the SLM (red upward triangles). Due to the low overhead for all the scaling costs, the total average duration per hologram does not significantly increase up to at least 2400 tweezers.

from GPU to CPU memory; and the time it takes to display the hologram on the SLM. In Fig. 6, we plot the average durations per step over all repetitions. The error bars are the standard deviation of the mean per sample. The average total time (purple pentagons) is 2.736(6) ms per computation and display cycle. We note that the measured timing may vary with different hardware.

The total average duration stays approximately constant because of two nearly constant bottlenecks in the current implementation of our method: the retrieval of the hologram from GPU to CPU memory (green squares), which takes 0.821(6) ms, and the display of the pattern on the SLM (red upward triangles), which takes about 1.772(15) ms. In the current implementation, the computation of the next pattern is halted until the SLM has finished updating its display, such that these durations add up directly to form the majority of the total duration. An improvement would be the parallelization of these tasks, so that the computation is not halted by the display process. Other future improvements include an improved resource sharing between GPU and CPU to reduce memory transfer times and the use of a faster refresh rate SLM [30–32].

The total duration of a rearrangement will thus be linearly dependent on the number of consecutive holograms necessary for the rearrangement. This number is determined by the longest trajectory of an atom. In the frequently studied case of rearranging square grids into another square grid with the same tweezer spacing, the longest potentially required trajectory scales proportional to $\sqrt{N_{tw}}$. Depending on the final tweezer pattern, one can take advantage of the versatility of SLMs and choose an initial tweezer pattern that has many reservoir tweezers next to the target tweezers [19,33], reducing the probable trajectory lengths and therefore rearrangement duration. Additionally, one can consider moving atoms over distances greater than a single Fourier unit, which could greatly reduce the total number of holograms. However, we leave this for future work.

The limit to how many atoms can be rearranged using this method ultimately comes from the loss rate per tweezer [9]. Treating the success rate per rearrangement cycle as an independent probability $p$ for every tweezer, the probability of a full defect-free arrangement scales with $N_{tw}$ as $p^{N_{tw}}$. In the experiments presented in Sec. 2 of this paper, the number of tweezers was limited to 36 by the available 813-nm laser power. For such a low tweezer number, a single rearrangement cycle proved to be sufficient to detect defect-free final geometries in 82.5 % of the realizations. For much larger arrays however, this will not be the case. For example, using the measured rearrangement success probability of p=0.997, the probability of successfully rearranging $N_{tw} = 1000$ atoms in tweezers is around 5 %. In the other 95 % of the cases, only a few atoms ($\leq 10$) are expected to be lost, and a second rearrangement cycle to fill the gaps can drastically increase the success rate. Simulations in reference [34] show that, for square geometries, more rearrangement cycles are beneficial when the losses from imaging and rearrangement are below $1/\sqrt{N_{tw}}$. For the values reported in this work, this is true for array sizes of up to thousands of tweezers. We note that such large arrays with sufficient trap depth on our current apparatus would require more than 100 W of 813-nm laser power, which is unrealistic. For other tweezer machines with fewer optical losses and a more favorable polarizability of the atomic species, thousands of tweezers can be obtained, and our technique could be employed [8, 10].

## 5 Discussion and outlook

Looking forward, the LPI method presented here shows promise for rearranging large tweezer arrays into arbitrary geometries. The reported success probability of rearrangement and imaging of >0.99 in this work is competitive with AOD sorting methods [1, 10, 14]. Under the assumption that fewer than a few tens of holograms are needed, the LPI method can rearrange atoms in tens of milliseconds. This is comparable to the total rearrangement duration with existing AOD sorting methods in arrays of a few hundred atoms [1, 2, 16]. A detailed comparison of total sorting time between the LPI method and AOD-based methods depends on the total number of holograms and the number of AOD tweezers that are used in parallel. For example, using eight holograms per lattice period, the current LPI implementation could rearrange the same arrays as [8] in $\sim 130\,\text{ms}$, which is an order of magnitude faster than their single-AOD demonstration. Assuming six holograms per lattice period and parallelizing the display of each hologram with the calculation and memory transfer of the next, the LPI method could assemble a $78 \times 78 \approx 6100$ square array in a similar time. In this case, the LPI method with a single SLM would match the performance of a technically much more complicated solution using multiple pairs of AODs with many tweezers per pair [10]. Moreover, using the SLM allows movements over the whole array, whereas commercially available AODs are lacking field-of-view for transport over such large arrays [10].

A potential improvement of the success probability of rearrangement could be artificially enlarging the computational space with zero-padding [19], so that atoms can move less than one Fourier unit per hologram. This improvement comes at the cost of increased computational time and number of holograms.

Besides the scalability of the LPI method, we foresee several other ways in which it increases parallel operation in current tweezer machines. In previous experiments with SLMs, tweezers that were not used after rearrangement remained on, essentially wasting optical power on empty tweezers. Having the capability to turn off unused tweezers makes the LPI method very efficient in the use of laser power. In machines that load tweezers continually such as those in references [7, 9], we expect that the availability of extra laser power after extinguishing empty tweezers can readily be useful: After using the SLM for rearrangement,

one could hold the desired atoms in a storage zone, while using the extra laser power to create and fill new tweezers in a loading zone.

Having real-time control of holograms on the SLM is also a promising tool for site-selectivity. To obtain site-selectivity in tweezer arrays one typically illuminates specific tweezers with non-magic trapping light, making use of a differential AC Stark shift [23, 35, 36]. Implementations using AODs cannot illuminate many atoms that are not in the same row or column. With a high-speed SLM that creates patterns of non-magic light, one can quickly update illumination patterns to select specific atoms.

Moving atoms using an SLM could also serve as a tool for coherent transport of qubits. Since moving a tweezer only requires modifying the hologram and not sweeping RF frequencies (which change the tweezer laser frequency), one can maintain magic trapping wavelength conditions. This could be particularly interesting for qubit encodings such as optical clock qubits [37] and fine structure qubits [38, 39].

Lastly, the use of an SLM allows for complex changes in tweezer geometry. Once a stochastically loaded sample has been rearranged, the position of every atom is known. Pre-calculated hologram sequences can then be used to further move or otherwise change tweezers, making it possible to use holograms that are beyond the scope of our fast atom sorting method. This opens up possibilities for 3D rearrangement [40] and could be interesting to create complex changes in the connectivity of Rydberg atom arrays [20, 22]. Decreasing the distance between atoms could be used to create strong Rydberg interactions between many atoms, while increasing the distance could be useful for imaging the atom arrays. With methods developed to calculate holograms that produce very tightly spaced tweezer arrays [41], a next step for the method would be to explore if atoms can be reversibly rearranged into such tightly spaced arrays using an ultra-high speed SLM.

# 6  Conclusion

We have demonstrated fast parallel rearrangement of single atoms in tweezers by displaying a sequence of holograms on an ultra-high speed SLM. The holograms were calculated using linear interpolation of tweezer positions and phases. This technique is capable of sorting many atoms into arbitrary geometries, regardless of the initial geometry. We also showed that multiple rearrangement cycles of the same atomic sample into different geometries are possible. We could update the tweezer positions every $2.736(6)$ ms, with several options for further speedup. This number does not vary significantly for arrays of up to at least 2400 tweezers and is mostly limited by technological restrictions. We expect only mild $\sqrt{N_{\text{tw}}}$ scaling for the total rearrangement time for typically used tweezer patterns. When correcting for losses due to imaging survival, the rearrangement success probability per atom was found to be $0.996(2)$. Combined with sufficiently high imaging survival, this opens the door to the fast assembly of thousands of single atoms in the near future.

# Acknowledgments

We thank T. Pfau, F. Meinert, M. Morgado, the TU/e neutral atom quantum computing team and Meadowlark Optics for discussions. We are grateful to Daniel Bonn for lending us a high-speed camera.

**Author contributions**  I.K. and Y.C.T. developed the method. I.K. created the implementation of the method. I.K., Y.C.T. and A.U. performed the measurements. I.K. analyzed the data. R.S. and F.S. supervised the work. All authors contributed to the manuscript.

**Funding information**    This work was supported by the Dutch National Growth Fund (NGF), as part of the Quantum Delta NL programme. It has also received funding under Horizon Europe programme HORIZON-CL4-2021-DIGITAL-EMERGING-01-30 via project 101070144 (EuRyQa). We thank QDNL and NWO for grant NGF.1623.23.025 ("Qudits in theory and experiment").

**Note**    During the completion of the manuscript, the authors became aware of the concurrent work presented in reference [42].

## A    Experimental details

Each experimental sequence described in this work starts by loading cold ($1\,\mu$K) atoms from a single frequency magneto-optical trap (MOT) operating on the $^1S_0 - ^3P_1$ transition at 689 nm into an array of optical tweezers and using light-assisted collisions to obtain single atoms [23]. The tweezers are formed by illuminating the ultra-high speed SLM with 450 mW of 813-nm light with a Gaussian beam waist of 6 mm. The SLM is operated at a chip temperature of 33 °C to minimize phase flicker, while retaining a fast update rate. On the SLM, a hologram is displayed that is the sum (modulo $2\pi$) of the intended target hologram, an aberration correction pattern, a blazed grating and a Fresnel lens phase, such that a real image of the tweezer pattern is formed approximately 1110 mm after the SLM. An achromatic doublet ($f = 500$ mm) and a microscope objective (Mitutoyo G Plan Apo 50X, $f = 4$ mm) image this inside the vacuum chamber to form the tweezers. The SLM and the achromatic doublet are 1680 mm apart. The distance between the achromatic doublet and the objective is 680 mm to achieve approximate aperture conjugation. Due to losses dominated by the absorption of the backplane of the SLM, only 235 mW of the 813-nm power is left before the microscope objective. We estimate that less than 40 % of this power is effectively used to form tweezer traps, mostly due to losses inside the objective, because the elements in the objective are not coated to be used at this wavelength. At maximum power, we estimate the trap depth by measuring differential AC Stark shifts on the $^1S_0 - ^3P_1$ ($m_J = \pm 1$) transition to be about 130 $\mu$K (270 $\mu$K) for the arrays containing 36 (16) atoms.

Imaging the atoms is done by recording the atomic fluorescence under illumination of 689-nm light, following the protocol explained in our previous work [23]. Each image in this work had a total duration of 200 ms, during which the intensity and frequency of the 689-nm light was alternating in 20 cycles between either favoring many scattering events ($I/I_s \approx 320$ during 88 % of the image) or cooling the atoms ($I/I_s \approx 70$ during 12 % of the image). The detuning varies with trap depth. We used $-720$ kHz ($-760$ kHz) and $-1420$ kHz ($-1460$ kHz) as imaging (cooling) frequencies for the 36- and 16-atom arrays, respectively. For each tweezer, we identify a region-of-interest (ROI) of $3 \times 3$ pixels in which we sum the collected fluorescence. To correct for observed drifts in the tweezer position we translate each ROI around 1 step in each direction and record the maximum fluorescence. We label an atom as being present if the signal is over a pre-set threshold value. Detection fidelities and imaging survival are discussed in Appendix B.

## B    Data analysis and fidelities

We characterize the detection fidelities and imaging survival for the $4 \times 4$ and $6 \times 6$ arrays. By taking more than 1000 images of atoms in either pattern, we obtain histograms of the collected fluorescence in each ROI. After fitting the zero-atom and single-atom peaks in the histograms

with an Erlang distribution and a skewed Gaussian distribution, we determine a binarization threshold [24,43,44]. We obtain a value for the true-negative fidelity ($F_0$) by looking at the fraction of the fitted probability distribution of the zero-atom peak that is below the threshold. Similarly, we obtain a true-positive fidelity ($F_1$) by looking at the fraction of the one-atom peak that is above the threshold.

To characterize the imaging survival, we take two consecutive images. We use the detection fidelities to get the probability $P(I_0)$ of detecting an atom in the first image, based on the probability $p_1$ that an atom was present in the tweezer:

$$P(I_0) = F_1 p_1 + (1 - F_0)(1 - p_1). \tag{B.1}$$

After the first image, the probability of an atom being still present is $S p_1$, where $S$ is the imaging survival. The probability of detecting an atom in both the second and the first image is given as:

$$P(I_1 \cap I_0) = F_1^2 S p_1 + F_1(1 - F_0)(1 - S)p_1 + (1 - F_0)^2(1 - p_1). \tag{B.2}$$

We measure the raw image survival $S_0$ as the conditional probability of detecting an atom in the second image, provided one was detected in the first image:

$$S_0 = \frac{P(I_1 \cap I_0)}{P(I_0)}. \tag{B.3}$$

Solving for $S$, we obtain:

$$\begin{aligned} S &= \frac{(S_0 + F_0 - 1)(F_1 p_1 + (1 - F_0)(1 - p_1))}{p_1 F_1(F_1 + F_0 - 1)} \\ &= \frac{S_0 + F_0 - 1}{F_1 + F_0 - 1}\left(1 + \frac{(1 - F_0)(1 - p_1)}{p_1 F_1}\right). \end{aligned} \tag{B.4}$$

We note that this expression differs from the one in reference [43]. Under the assumption that $p_1 = 1$, the results become the same. This would neglect the stochastic loading of atoms, which we find unjustified. It should be noted that $p_1$ and $S$ are independent variables, and $S_0 = S_0(S, p_1)$ such that $S$ in Eqn. (B.4) will always yield $S \leq 1$ and an apparent divergence as $p_1 \to 0$ never occurs.

Assuming $p_1 \approx 0.45$, the results of the characterization measurements are summarized in Tab. 1. We observe that the image survival probability is higher for deeper traps. In line with our previous work, we suspect this is due to lower axial trapping frequencies in shallower traps that ultimately limit the Sisyphus cooling. We observed that for 16-atom patterns other than the $4 \times 4$ array the fidelities were similar, again indicating that the trap depth is the important parameter for the values for the fidelities. For simplicity of the next derivation, we assume the exact same values for all 16-atom patterns.

Next, we use the above fidelities to derive corrected values of the rearrangement success probability. In the first rearrangement experiments presented in Sec. 2, atoms were rearranged from a $6 \times 6$ array to a $4 \times 4$ array. Because the detection fidelities and imaging survival depend on the trap depth, we adopt subscripts to denote the array size. Using the same procedure as above, the probability $R$ for an atom to survive the rearrangement can be expressed in terms of the measured fraction $R_0$ as:

$$R = \frac{(R_0 + F_{0,16} - 1)(F_{1,36} p_1 + (1 - F_{0,36})(1 - p_1))}{S_{36} F_{1,36} p_1 (F_{1,16} + F_{0,16} - 1)}. \tag{B.5}$$

Taking $R_0 = 0.988(4)$ from the main text, a corrected value of $R = 0.997^{+0.003}_{-0.006}$ is obtained using $p_1 \approx 0.45$ and the detection fidelities from Table 1. We put an upper limit to the uncertainty to avoid non-physical values of $R > 1$.

Table 1: Average detection fidelities and survival probabilities for the $4 \times 4$ and $6 \times 6$ arrays. Values are averaged mean values over all tweezers and the errors are standard deviations of the mean.

| Array Size | Trap Depth | $F_0$ | $F_1$ | $S_0$ | $S$ |
|---|---|---|---|---|---|
| 36 | $130 \, \mu\text{K}$ | 0.9986(13) | 0.997(3) | 0.988(3) | 0.993(5) |
| 16 | $270 \, \mu\text{K}$ | 0.9992(7) | 0.9998(2) | 0.9966(13) | 0.9978(16) |

The result above can be extended to $n$ rearrangements by assuming every rearrangement has the same success probability and every 16-atom pattern shares the same fidelities. We note that especially the first assumption is not obvious, since the number of holograms per rearrangement cycle in the experiments in Sec. 2 were not constant. Using these assumptions, the corrected rearrangement success $R$ based on a result $R_0$ is given as:

$$R = \left( \frac{(R_0 + F_{0,16} - 1)(F_{1,36} p_1 + (1 - F_{0,36})(1 - p_1))}{S_{16}^{n-1} S_{36} F_{1,36} p_1 (F_{1,16} + F_{0,16} - 1)} \right)^{1/n} . \tag{B.6}$$

Using $n = 4$ cycles and $R_0 = 0.968(7)$, we obtain $R = 0.996(2)$ as the success probability for a single rearrangement, excluding imaging survival. This matches very well with the above result for $n = 1$.

## C Derivation of phase slip per displacement

Here, we present a derivation for the amount of tweezer phase slip $\xi$ acquired per move for a single spot due to a displacement $d$ of the real optical axis compared to the assumed optical axis. The electric field in the focal plane of the objective is given by the discrete Fourier transform:

$$\mathbf{U}(m\Delta x', n\Delta y') = \sum_{k=-\frac{N_x}{2}}^{\frac{N_x}{2}} \sum_{l=-\frac{N_y}{2}}^{\frac{N_y}{2}} |U(k\Delta x, l\Delta y)| e^{i2\pi\left(\frac{(k-d_x)m}{N_x} + \frac{(l-d_y)n}{N_y}\right)} e^{i\varphi(k,l)} , \tag{C.1}$$

where $U$ is the profile of the illuminating tweezer laser. For ease of computation, we assume that the profile of the illuminating laser is uniform ($U = 1$). The pixel indices are $(k, l)$ in the SLM plane and $(m, n)$ in the Fourier plane. $\varphi(k, l)$ is the hologram displayed by the SLM with $N_x \times N_y$ pixels. The pixel spacing is given by $(\Delta x, \Delta y)$, which leads to Fourier units $(\Delta x', \Delta y') = \left( \frac{\lambda f}{L_x}, \frac{\lambda f}{L_y} \right)$, with $\lambda$, $f$ and $L_i$ the laser wavelength, the objective focal length and demagnified size of the SLM at the back focal plane of the objective, respectively. We account for a possible displacement of the optical axis with respect to the SLM's center pixel by introducing a relative coordinate shift $(d_x, d_y)$ between the exponent of the Fourier transform and the hologram.

In the ideal situation the optical axis aligns with the SLM's center, so that $d_x = d_y = 0$. To form a single tweezer, no hologram is needed. If we substitute $\varphi = 0$ into Eqn. (C.1), we find that for sufficiently large $N_x$ and $N_y$ the formed pattern approximates up to some constants a Dirac delta function: $\mathbf{U}(m\Delta x', n\Delta y') \propto \delta(m = 0, n = 0)$. Moving the position of the single tweezer can be done by choosing a phase gradient as a hologram. For example, a move of one

Fourier step in the $x$-direction can be done with a hologram $\varphi(k,l) = 2\pi k/N_x$. Substituting this into Eqn. (C.1) results again in an approximated delta function, but now translated to $\delta(m=-1,n=0)$. We define the "computational center" of the hologram as the position at which it remains unchanged when moving tweezers, which is the central pixel for our specific choice of phase gradient.

Next we look at what happens when there is a displacement $(d_x, d_y)$ between the computational center of the hologram and the optical axis. By taking the displacement term out of the summation, we see that the effect of the displacement is a coordinate dependent phase shift of the optical phase of the tweezer of $-2\pi\left(\frac{md_x}{N_x} + \frac{nd_y}{N_y}\right)$. Every move of a tweezer is the change of a coordinate in Fourier units by 1 (e.g. $m \mapsto m-1$). Substituting both coordinates into Eqn. (C.1) yields a tweezer phase shift of $\xi(d_x) = 2\pi\frac{d_x}{N_x}$ per move. Note that for the shifted hologram data shown in Fig.5a we shifted the hologram with respect to the SLM by 250 pixels using circular boundary conditions instead of moving the SLM.

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
