# Peer review of "Parallel assembly of neutral atom arrays with an SLM using linear phase interpolation"

_SciPost Physics, doi:SciPost Phys. 19, 118 (2025)_

## Round 1 · Referee Report · Anonymous (Referee 1) · 2025-3-4

Strengths

1) Very well written and explaining the method carefully 2) Very impressive results for 16 atoms 3) Good description on how suppressing phase fluctuation is key to their success

Weaknesses

1) Lack of experimental details that allow the readers to assess the applicability of the method 2) Extrapolation of the results to thousands of atoms, without providing sufficient details how the extrapolation is being carried out 3) No explanation about the limit to their success (0.988)

Report

The paper describes the use of an SLM to trap single atoms in optical tweezers. Unlike using AOM's to trap the atoms, the use of the SLM allows the authors to simultaneously move all the atoms at the same time from their initial to their final position. This can severely decrease the time necessary to perform the transfer and thus is very convenient for larger number of atoms. Although they use the method to transfer only 16 atoms probably due to limited power in the trapping laser, the method could be extended to very many atoms and thus is very interesting from a viewpoint of quantum computation and quantum simulation.

The paper is very well written, and discusses various steps in the research. However, some details are missing from the paper and it is not clear how various conclusions by the authors can be verified. Below I will mention those issues. Overall, I think the paper can be published by taking these recommendations into account. Since the paper addresses an interesting topic in the field of atomic manipulations for quantum computing and simulation, the journal SciPost Physics is appropriate.

Requested changes

My concerns:

1) Although the paper is based on experiments described in Sec. 2, there is not much experimental details. For instance, there is no overview of the experimental setup. Referring to a previous paper does not help, since no detailed overview of the setup is provided in that paper. Also, powers used for the trapping and imaging are not discussed, although the power seemed to be limiting.

2) The average filling fraction for 16 atoms is 0.988, which is a big achievement. However, it is not discussed at all, what limits this filling fraction. Is this a fundamental limit of the technique, is this caused by the limited power, or are there other technical reasons, why it is this number.

3) The final sentence of Sec. 2 concludes "This corresponds to a success rate of ..". It is not clear, where this number (0.991) is based on, and how it scales with the number of atoms (16) involved. And again, what limits this number? And how why does it make the method suitable for adjusting geometries on the same atomic sample?

4) In Sec. 4 it is concluded that the method is nearly independent of Ntw. Figure 6 is an illustration of it. However, there are certain systematic effects visible that are not discussed. For instance, apparently the computation of the hologram takes more time for a small number of Ntw. Also, the display on SLM seems to go faster for more Ntw. And the error bar in the total time increases significantly for more than 2000 Ntw. These effect are all small, but require some interpretation from the authors to get a better feeling on the scaling with Ntw.

5) The final paragraph in Sec. 4 is very qualitative, and not quantitative. The values for 36 tweezers are extrapolated to many thousands of atoms, but unclear is how the scaling can be trusted. What kind of powers are needed for 1000 atoms? What success probability is allowed for scaling up to 1000 atoms? When will multiple rearrangements cycles becomes necessary? Of course, as with any extrapolation there are uncertainties, but to get a better feeling on how the method will work for 1000 atoms, more information is required.

6) The abstract and the conclusion mention a specific number for the update time, namely 2.72(2) ms. This number seems to be very depending on the equipment that the authors use, and will be different from setup to setup. Given the very concrete value that the authors use, it looks like a generic result for the technique. Perhaps toning this statement a bit down to a few ms, creates a better impression of the capabilities of the method.

Recommendation

Ask for minor revision

  • validity: high
  • significance: high
  • originality: good
  • clarity: top
  • formatting: excellent
  • grammar: good

Author:  Ivo Knottnerus  on 2025-04-17  [id 5382]

(in reply to Report 1 on 2025-03-04)
Category:
answer to question
correction

Dear referee,

We are very grateful for your feedback and for the concerns you raised. Your report highlighted several places where the paper could be improved, and have tried and adjusted the manuscript to take away your concerns. We believe the work is stronger after implementing this feedback, and hope you agree. A list of all changes (including some typos and other minor changes) is presented as accompaniment to the resubmission. Please find attached an elaborate motivation for the changes based on your concerns.

Best regards,
Ivo

Attachment:

Response_Ref1.pdf

---

## Round 3 · Referee Report · Anonymous (Referee 3) · 2025-6-5

Strengths
1- SLM re-arrangement has important implications for preparing atomic arrays as it is expected to scale with atoms numbers. This work demonstrated a great improvement from previous achievements, with the smart, simple and clearly explained idea of linear interpolation of the tweezers phase.
2- It was a pleasure to read the paper and it should become a must-read for anyone working with holographic arrays of tweezers.
Weaknesses
1- A study of the minimal step size required to successfully move the atom would have been a nice addition to the presented results.
Report
The core idea of “linear phase interpolation” is very well explained (one of my master-level student could reproduce the work based on the paper explanation), and its importance is effectively demonstrated in Figure 4. The more subtle point of aligning the phase mask “center” and optical axis is also nicely highlighted in Figure 5.
The potential scalability to 1000s tweezers is well supported in Figure 6.
It would have been interesting to have a more detailed study of how to choose the minimal step size when moving a tweezers from one spot to the next. This could be left for future work though, as the main point of this paper is rather to point the importance of controlling the tweezers phase during displacement.
Overall, the work is a significant technical improvement, and I enjoyed much reading the paper as the underlying ideas are clearly conveyed to the reader.
The paper should definitely be published in SciPost Physics following the minor modification requested.
Requested changes
(1) I would like to read the following numbers in the paper: the size of the tweezers and the minimum step size (the expression lambda*f/mL is given in the paper, but one cannot compute it as L is not given). I feel they are important for the reader to visualize quantitatively how small are the changes between 2 holograms. Currently, only a qualitative comment is given (“overlap between consecutive tweezers”).
(2) Is there laser-cooling during the re-arrangement? If yes, this should be mentioned explicitly. If not, could this help to make larger steps and thus reduce the re-arrangement time?
(3) The re-arrangement time is dominated by the SLM refresh rate, but 30% is from the memory transfer. Could the authors comment briefly if this could be improved by implementing Direct Memory Transfer between the GPU and the SLM (bypassing the CPU)? What is the limit set by the PCIe bus bandwidth?
Recommendation
Publish (easily meets expectations and criteria for this Journal; among top 50%)
Dear Referee,
Thank you for the positive words and suggestions. We have taken your requested changes into consideration and adjusted the manuscript accordingly. We have implemented the first point and explicitly state the sizes. Furthermore, we do not use laser cooling, for reasons explained in the accompanying document, but we nevertheless think that larger steps are possible. A detailed investigation would require significant effort, so we leave it to further work. Finally, we provide in the attached document some calculations on the final limits with the current hardware.
We hope that these adjustments satisfy your requests.
Best regards, on behalf of the authors,
Ivo Knottnerus

Ivo Knottnerus on 2025-04-17 [id 5384]
Dear Editor,
Please find attached also a redlined version of the manuscript, where the changes are marked in red and where possible, text from the previous version is striped out. Note that this version does not have figures, because it was otherwise too large for the upload in the file attachment.
Best Regards,
Ivo
Attachment:
250414_SLM_Sorting_wChanges.pdf
Ivo Knottnerus on 2025-04-17 [id 5383]
Dear Editor,
Please find attached the list of changes to the manuscript. A more extensive motivation on the changes based on the referee report on the first version is attached to a reply to their report. Thank you for your work on our manuscript. We look forward to hearing from you.
Best Regards,
Ivo
Attachment:
Changes_to_the_manuscript.pdf

---

## Round 3 · Referee Report · Anonymous (Referee 4) · 2025-6-9

Report
by Ivo H. A. Knottnerus, Yu Chih Tseng, Alexander Urech, Robert J. C. Spreeuw, and Florian Schreck
This study introduces a new method for the rapid rearrangement of neutral atom arrays using a spatial light modulator (SLM), leveraging direct Fourier transform calculations that account for both the intensity and optical phase of each tweezer. Compared to conventional approaches based on the Gerchberg–Saxton (GS) algorithm, this technique achieves significantly improved computational speed and rearrangement success rates. In particular, the method shares conceptual similarities with the concurrently developed work “AI-Enabled Rapid Assembly of Thousands of Defect-Free Neutral Atom Arrays with Constant-Time-Overhead” by the USTC group (Ref. [42], cited on p.11), and demonstrates experimental performance on arrays of up to 36 traps, with simulation benchmarks showing constant computational times even for systems exceeding 2400 tweezers.
The proposed method effectively overcomes key limitations of GS-based SLM rearrangement, particularly by reducing computational time and minimizing atom loss. This advancement facilitates efficient large-scale atom array rearrangement without incurring additional time overhead. Leveraging the inherent parallelism of SLMs, the approach presents a promising alternative to AOD-based techniques, particularly in high-density configurations. The authors further investigate the role of optical phase by applying linear interpolation between initial and final configurations, and assess success probabilities as a function of frame intervals—thereby identifying key factors contributing to intensity flicker and atom loss.
In that regards, we conclude that this manuscript is worthy of being published with some improvements suggested below:
Comments and Suggestions:
(1) A quantitative benchmark would strengthen the manuscript—for instance, identifying the array size or regime at which the proposed method surpasses AOD-based rearrangement in transport speed. A direct performance comparison with the GS algorithm would also be valuable in demonstrating the method’s practical benefits.
(2) In Figure 3, it would be helpful to include visual markers for trap and atom positions, similar to those in Figure 2, to enhance clarity and consistency.
(3) The argument in Chapter 3 and Figure 4(b) regarding atom loss due to phase differences could be further substantiated by including plots of the inter-frame phase differences. Correlating these with the intensity flicker observed in Figure 4(a) would reinforce the proposed interpretation.
Recommendation
Ask for major revision
Dear Referee,
Thank you for the great suggestions. We have taken them into consideration and adjusted the manuscript accordingly. Specifically, we added a more quantitative discussion on the scaling arguments presented in the discussion. As you can see in the attached document, a detailed discussion would require a lengthy discussion. We therefore chose to give a comparison with two recent examples. Furthermore, we added the ROIs in Fig.3 as per your request. Regarding the correlation, we found ourselves unable to make a quantitative correlation in Fig.4, as further elaborated on in the attached document.
We hope that these adjustments address your concerns appropriately.
Best regards, on behalf of the authors,
Ivo Knottnerus
Attachment:

---

## Round 3 · Author Response

probability of imaging and rearrangement. This gives a better understanding of the low loss observed from this rearrangement method and helps in extrapolations to other experiments or larger arrays. In that light, we have also quantified the paragraph on scaling
up to larger arrays. Last, we redid the benchmarking with more data points to remove artifacts in the results and more clearly show the nearly constant scaling with number of tweezers for the hologram calculation.

---

## Round 3 · List of Changes

-
Added Appendix A. Experimental Details;
-
Added Appendix B. Data Analysis;
-
Changed from reporting the imaging and rearrangement success to an isolated rearrangement success probability as calculated with the formulas presented in the appendix on the data analysis. Added corrected rearrangement success probabilities at all relevant place;
-
Quantified better the paragraph on scaling up by including a small example of the expected success rate in a first rearrangement for thousands of atoms. Also included some numbers for why this is not realistic on our apparatus with 813-nm light;
-
Reran the benchmarking that is presented to figure 6. By taking care that the computer was not running any other processes, repeating the run in different orders and including more data points, most of the artifacts in the old data have been removed. Updated the numbers with the results;
-
Rewritten the expression for the Fourier unit such that it is clearer what the magnified length of the SLM means. Corrected the value of the magnification as well to 0.41 instead of 0.31;
-
Corrected other typos (most notably: the SLM product number);
-
Added references to relevant papers.

---

## Editorial Decision

published